# Medication Adherence in Medicare-Enrolled Older Adults with Chronic Obstructive Pulmonary Disease before and during the COVID-19 Pandemic

**DOI:** 10.3390/jcm11236985

**Published:** 2022-11-26

**Authors:** Ligang Liu, Armando Silva Almodóvar, Milap C. Nahata

**Affiliations:** 1Institute of Therapeutic Innovations and Outcomes (ITIO), College of Pharmacy, The Ohio State University, Columbus, OH 43210, USA; 2College of Medicine, The Ohio State University, Columbus, OH 43210, USA

**Keywords:** COPD, Medicare, medication adherence, COVID-19 pandemic, geriatric

## Abstract

Medication adherence to controller inhalers was unknown in older Medicare patients with chronic obstructive pulmonary disease (COPD) before and during the pandemic. This study evaluated changes in medication adherence to controller medications and factors associated with high adherence. This retrospective cohort study included older Medicare patients with COPD. The proportion of days covered (PDC) reflected changes in medication adherence from January to July in 2019 and in 2020. Paired *t*-test evaluated changes in adherence. Logistic regression determined the association of patient characteristics with high adherence (PDC ≥ 80%). Mean adherence decreased (*p* < 0.001) for long-acting beta-agonists, long-acting muscarinic antagonists, and inhaled corticosteroids in 2020. The percentage of patients with high adherence dropped from 74.4% to 58.1% (*p* < 0.001). The number of controllers, having ≥3 albuterol fills, and a 90-day supply were associated with high adherence in 2019 and 2020 (*p* < 0.001). The COVID-19 pandemic may negatively impact medication adherence. Patients with evidence of more severe diseases and a 90-day supply were more likely to adhere to therapy. Healthcare professionals should prioritize prescribing 90-day supplies of medications and monitor drug-related problems as components of pharmacovigilance to enhance adherence to therapies and the desired clinical outcomes among patients with COPD.

## 1. Introduction

Chronic obstructive pulmonary disease (COPD) and other chronic lower respiratory diseases were the sixth leading causes of death in 2020 [1]. These illnesses affected over 250 million people around the world, causing substantial burdens on healthcare systems [2]. Common comorbidities of COPD have included skeletal muscle dysfunction, cardiovascular disease, mental health disorders, and lung cancer [3,4,5,6].

Pharmacological therapies for patients with COPD include short-acting beta agonist(s) (SABA), short-acting muscarinic antagonist(s) (SAMA), long-acting muscarinic antagonist(s) (LAMA), long-acting β2-agonist(s) (LABA), and inhaled corticosteroid(s) (ICS) [7]. Adherence to guideline-directed medical therapy (GDMT) for COPD may improve clinical outcomes and reduce all-cause and respiratory-specific emergency department visits and hospitalizations [8,9].

Patient-specific factors, including mental health diagnoses (particularly depression), lower health literacy, medication-related problems, and lower sociodemographic status, have been associated with decreased medication adherence, increased readmission rates for acute COPD exacerbations, and poorer health outcomes [10,11]. Older adults are vulnerable to medication-related problems, and services such as comprehensive medication reviews need to be provided to improve adherence [12].

The COVID-19 pandemic limited access to healthcare for COPD patients [13]. Kahnert et al. found a deterioration of clinical status in the COPD population during the pandemic [14]. The GOLD Science Committee also emphasized that the COVID-19 pandemic had made COPD management challenges and urged actions to improve care for patients with COPD during the pandemic [15].

Conflicting data exist on medication adherence among patients with COPD. Several studies found increased medication adherence in older patients with asthma and/or COPD after the beginning of the pandemic [16,17]. However, Barrett et al. reported decreased prescription claims in COPD and asthma patients compared to the pre-pandemic time [18]. Zhang et al. observed similar adherence during the pandemic compared to the pre-pandemic period in the general COPD population [19]. Importantly, there are no data on medication adherence in Medicare-enrolled older adults with COPD alone from the COVID pandemic period.

The objectives of this study were to (1) examine medication adherence in older adults with COPD, (2) determine the impact of the COVID-19 pandemic on medication prescribing patterns and medication adherence in this population, and (3) evaluate the associations of specific sociodemographic and patient characteristics with medication adherence in older adults in the presence of the COVID-19 pandemic.

## 2. Materials and Methods

### 2.1. Study Design

This was a retrospective longitudinal cohort study of older Medicare patients with COPD. Eligible patients were individuals enrolled in medication therapy management (MTM) services, were diagnosed with COPD, and had at least two prescription claims for the same controller medication in 2019. Patients with cystic fibrosis or asthma or <65 years of age were excluded from this study. Medications included in this study were SABA, LABAs, LAMAs, and ICSs. Biologic medications were not included given they are largely covered under Medicare Part B; the MTM program did not have access to these data. The institutional review board at the Ohio State University approved this study (22 September 2020; study ID: 2020H0393).

### 2.2. Data Sources

Data were obtained from the MTM provider with permission from the insurance plan. These included the patient’s age, sex, zip code, international classification of disease, tenth revision (ICD-10) codes, number of medications, number of prescribers, number of pharmacies, prescription claims data, medication-related problems identified by electronic review of prescription claims, and proportion of days covered (PDC) for inhalers for this study.

Medication refill histories based on pharmacy claim data were used to measure patient adherence to maintenance inhalers at a fixed time from 1 January to 31 July 2019, and 1 January to 31 July 2020. The proportion of days covered (PDC) was used to calculate a patient’s adherence. The calculation for PDC equaled the number of covered days with a targeted medication divided by the number of days in a period. The tracking period started with the patient’s first fill during the observation period. If a PDC could not be calculated, it was assumed that the patient had a PDC of 0%. Patients with a PDC below 80% were considered non-adherent, and greater or equal to 80% were considered highly adherent. Appendix A shows a detailed list of medications assessed in this study.

### 2.3. Statistical Analysis

Microsoft Excel (Version 2209 Build 16.0.15629.20152) and IBM SPSS (version 28) were used to organize and analyze data. Nominal data were described by counts and percentages. Continuous data were presented by means and standard deviations (SDs). Age, prescribers, pharmacies, inhalers, and corticosteroids were transformed into ordinary variables.

Paired *t*-tests were used to assess differences in adherence to the same medication between 2019 and 2020. The percentage of patients adherent to medications in each period was reported and compared using McNemar’s test.

Exploratory logistic regression was used to identify potential predictors of medication adherence in 2019 and 2020 separately. Variables used in this regression included age, sex, number of prescriptions, number of rescue inhalers, number of medication-related problems, the diagnosis of depression, number of pharmacies, number of prescribers, number of oral corticosteroids, and having a 90-day medication supply. The analysis in the regression for 2020 included a variable that reflected whether the patients were adherent to the inhalers in 2019. Bonferroni corrections were used to provide a conservative *p* value that would establish significance.

## 3. Results

A total of 1533 patients were included in this study. The mean patients’ age was 76.14 ± 6.74 years, with 59% being female. They had 6.24 ±3.38 prescribers, were prescribed 13.71 ± 4.70 medications, and went to 2.48 ± 1.46 pharmacies to fill their prescriptions in 2019. In this cohort, 32.2% of patients received oral corticosteroids, and 16.9% had depression. 77.9% of the patients needed more than one controller medication for COPD. Detailed information can be found in Table 1.

About two-thirds of patients were highly adherent to their inhalers (LAMA [69.0%], LABA [66.8%], and ICS (65.9%]) in 2019. In the first several months of the pandemic, the average adherence rate to LABA [50.1%] and ICS [48.5%] decreased, and LAMA [69.1%] adherence did not change (Figure 1). Mean PDC for controller inhalers decreased significantly in 2020 compared with 2019(LABA, 83.52 ± 20.09 vs. 58.36 ± 40.82; LAMA, 84.25 ± 19.88 vs. 59.76 ± 41.91; ICS 82.99 ± 20.47 vs. 56.60 ± 41.15 (all *p* values < 0.001) (Figure 2). The proportion of patients with an ICS-LABA, and a LABA-LAMA combination inhaler decreased; however, the proportion of patients having an ICS-LABA-LAMA combination inhaler decreased (Figure 3).

The percentage of patients adhering to their treatment dropped from 74.4% in 2019 to 58.1% in 2020, with *p* < 0.001. The percentage of patients who received ICS and LABA decreased from 68% to 59%. However, the number of patients who used the combination of ICS, LABA, and LAMA increased from 71 to 91. The number of patients receiving oral corticosteroids decreased from 493 (32.2%) in 2019 to 414 (25.7%) in 2020.

In the regression model of 2019, the full model containing all predictors was statistically significant, Χ^2^ (27, N = 1533) =240.19, *p* < 0.001. The model as a whole explained between 14.5% (Cox and Snell R square) and 21.4% (Nagelkerke R squared) of the variance in sleep status, and correctly classified 77.3% of cases; variables that were associated with high adherence for controller inhalers were the number of maintenance inhalers (*p* < 0.001), having ≥3 albuterol inhalers (odds ratio [OR], 2.25; 95% confidence interval [CI], 1.74–2.91; *p* < 0.001), and a 90-day supply of controller medications (OR, 2.57; 95% CI, 1.88–3.53; *p* < 0.001). Patients with ≥3 controller medication classes had 2.89 to 6.44 times the odds of being adherent to COPD controller medications compared with patients with 1 controller (*p* < 0.001). A complete list of nonsignificant variables appears in Table 1.

In 2020, the full model containing all predictors was statistically significant, Χ^2^ (28, N = 1533) = 182.45, *p* < 0.001. The model as a whole explained between 14.3% (Cox and Snell R square) and 21.3% (Nagelkerke R squared) of the variance in sleep status, and correctly classified 77.5% of cases. Variables related to high adherence to inhalers included the number of controller classes (*p* < 0.001), having ≥3 albuterol medication (OR, 2.44; 95% CI, 1.73–3.46; *p* < 0.001), a 90-day supply of controllers (OR, 3.06; 95% CI, 2.34–4.00; *p* < 0.001), and being adherent to controller medications in 2019 (OR, 2.24; 95% CI, 1.63–3.08; *p* < 0.001). A complete list of nonsignificant variables is provided in Table 2.

## 4. Discussion

This was the first study to measure medication adherence and its changes during the COVID-19 pandemic compared to a pre-pandemic period using data from Medicare-enrolled, MTM-eligible older patients with COPD. We found a remarkable reduction in medication adherence to controller inhalers for COPD in the first several months of 2020. Significant indicators of adherence were patients receiving a 90-day supply of controller inhalers, ≥3 rescue inhalers, and the number of maintenance inhalers. Our findings suggested that access to medications and healthcare may have been disturbed in the pandemic’s first few months given the observed decreases in medication claims.

Our data demonstrated that the percentage of patients adhering to maintenance controllers was suboptimal in 2019 and 2020. The proportion of patients who had high adherence, as measured by PDC, ranged from 65.9% to 69.0% in 2019 and 48.5% to 69.1% in 2020. Nishi et al. also used the PDC to measure the adherence to LABAs, LAMAs, and ICSs in Medicare-enrolled older patients with COPD from 2008 to 2013, and found that mean adherence to maintenance medication was about 55% [20]. Moreover, another study discovered that 69.8% of existing diagnosed and 84.4% of new COPD patients were non-adherent to maintenance therapy, which was defined as PDC < 80%, using Medicare real-world data from 2007 to 2014 [21]. In general, adherence to maintenance medication was still suboptimal in Medicare-enrolled MTM-eligible older adults with COPD before and during the COVID-19 pandemic. It is important to recognize barriers to adherence, and additional measures must be taken to improve medication adherence, especially during an event such as a pandemic.

This study importantly discovered that patients with a 90-day supply were more likely to be considered adherent. During the pandemic, access to medications may have been impeded in older people with chronic conditions [22]. Ismail et al. observed that about 20% of patients with chronic diseases had trouble obtaining medications during the pandemic [23]. It was assumed that patients who made regular visits to pharmacies to obtain refills were also at an increased risk of exposure to infections [24]. Vordenberg et al. reported that over one-half of older adults continued to go to the pharmacy for medicines despite the risk of infection [25]. A 90-day supply of medications would reduce the frequency of visits to the pharmacy and the risk of exposure to the virus. Several studies have reported that patients with a 90-day supply were more adherent to medications compared to patients with a non-90-day supply [26,27]. The Department of Health and Human Services also advocated for a 90-day supply of medications during the pandemic [28]. Therefore, a 90-day supply of medications should be prioritized to optimize the therapy during the pandemic in patients with chronic diseases, including COPD.

It is observed in this cohort that patients with a higher number of controller medications were more likely to be adherent to their medications. This finding supported the health belief model of behavior that patients’ knowledge and perceptions of the disease and treatment were associated with good adherence [29]. Notably, patients adhering to medications in 2019 were more likely to adhere to medications in 2020, suggesting that evidence of previous medication adherence was an important predictor for adherence in the future [30].

It was found that 57.8% of patients in 2019 and 52.4% in 2020 had ≥3 rescue fills, possibly due to the patients being less active during the pandemic. This number may indicate the overutilization of rescue inhalers to control symptoms related to COPD, with implications of increased disease severity and high COPD burden [31,32]. Overuse of SABA may cause bronchodilator tolerance and decrease the response to rescue beta-agonist treatments [33]. Our study also discovered that oral corticosteroid fill claims decreased during the first few months of 2020. It is also reported that the number of patients with a moderate or severe exacerbation of COPD decreased during the pandemic [34], which may be explained by the use of face masks and other social distancing policies during the pandemic [35].

Older adults are more likely to experience medication-related problems due to polypharmacy and decreased liver and kidney functions [36], contributing to up to 30% of hospitalizations in the geriatric population [37]. In our study, we found patients with fewer medication-related problems were more likely to have high adherence, even though the association was weak. All patients included in this study were provided with MTM services, as a component of pharmacovigilance efforts made by pharmacists to identify and minimize medication nonadherence and maximize medication safety by detecting, assessing, and resolving drug-related problems to achieve desired clinical outcomes [38].

## 5. Limitations

This study only included Medicare-enrolled–MTM-eligible patients from one insurance company. Therefore, it may not represent the entire Medicare population. Further, this study only examined prescription claims in the first seven months of 2020 and did not capture the medication adherence changes during the entire pandemic period. Future research needs to be conducted to observe adherence trends amid the pandemic. Lastly, the status of the COVID-19 infection was not recorded in the system, and we were unable to assess the effects of a COVID-19 infection on adherence. The authors were unable to determine if providers discontinued medication therapies and could not assess the influence of cash claims that were not processed by the insurance. The authors were unable to assess historical trends of adherence prior to 2019 in this analysis.

## 6. Conclusions

The adherence to controller inhalers was suboptimal in Medicare–MTM-eligible older adults with COPD. Our study demonstrated that the COVID-19 pandemic might have had a negative impact on medication adherence during the initial months. Patients with signs of severe disease as evidenced by the receipt of a greater number of albuterol inhalers and more controller inhalers, and a 90-day supply, were more likely to adhere to the inhalers. Healthcare professionals should prioritize prescribing 90-day supplies of medications and monitor medication-related problems as components of pharmacovigilance efforts to achieve high adherence and desired clinical outcomes for the optimal care of COPD patients.

## Figures and Tables

**Figure 1 jcm-11-06985-f001:**
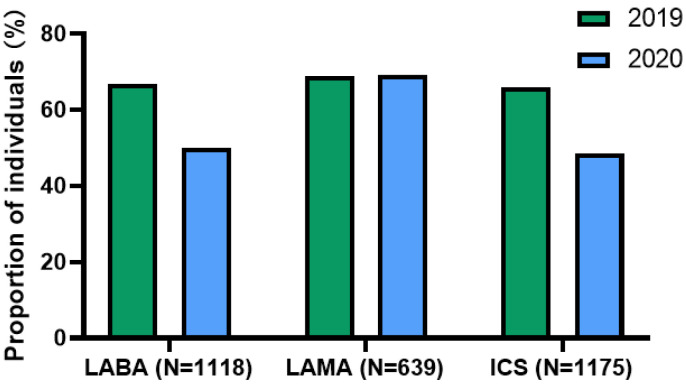
Proportion of individuals with high adherence to their controllers. Abbreviations: LABA, long-acting β2-agonists; LAMA, long-acting muscarinic antagonists; ICS, inhaled corticosteroid.

**Figure 2 jcm-11-06985-f002:**
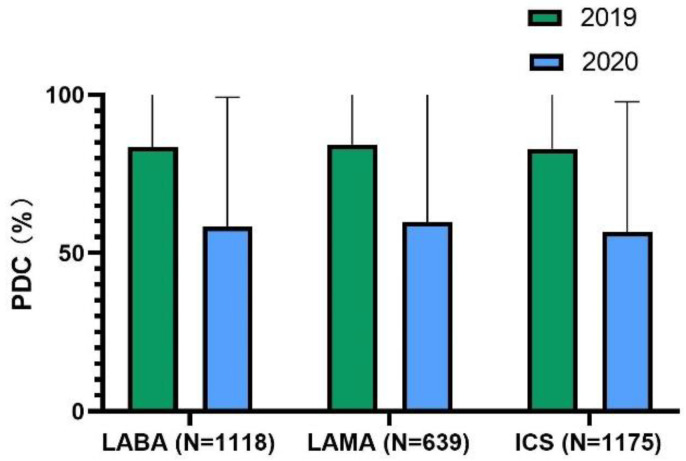
Mean proportion of days covered for controller inhalers. Abbreviations: LABA, long-acting β2-agonists; LAMA, long-acting muscarinic antagonists; ICS, inhaled corticosteroid.

**Figure 3 jcm-11-06985-f003:**
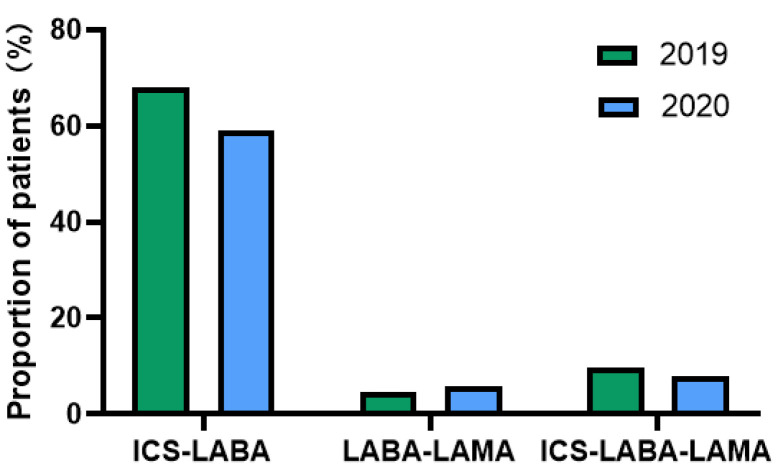
Proportion of patients with at least 1 combination inhaler. Abbreviations: LABA, long-acting β2-agonists; LAMA, long-acting muscarinic antagonists; ICS, inhaled corticosteroid.

**Table 1 jcm-11-06985-t001:** Descriptive data and results from logistic regression evaluating the association between patient characteristics and adherence to controller inhalers in 2019 (N = 1533).

Characteristic	Overall Cohort	With High Adherence	With Nonadherence	*p* Value	B Value	Adjusted Odds Ratio
	N = 1533	N = 1141	N = 392			(95% confidence interval)
	N (%)	N (%)	N (%)			
Age, year				0.36		
65–74 (reference)	644 (42.0)	484 (42.4)	160 (40.8)			
75–84	656 (42.8)	496 (43.5)	160 (40.8)	0.81	0.03	1.04 (0.78–1.37)
≥85	233 (15.2)	161 (14.1)	72 (18.4)	0.23	−0.23	0.80 (0.55–1.15)
Sex						
Male (reference)	629 (41.0)	466 (40.8)	163 (41.6)			
Female	904 (59.0)	675 (59.2)	229 (58.4)	0.51	−0.09	0.509 (0.71–1.19)
Number of medications				0.31		
8–10 (reference)	456 (29.7)	315 (27.6)	141 (36.0)			
11–13	400 (26.1)	293 (25.7)	107 (27.3)	0.82	0.04	1.04 (0.75–1.44)
14–16	299 (19.5)	228 (20.0)	71 (18.1)	0.51	0.13	1.14 (0.78–1.65)
17–19	200 (13.0)	159 (13.9)	41 (10.5)	0.08	0.39	1.48 (0.95–2.31)
≥20	178 (11.6)	146 (12.8)	32 (8.2)	0.11	0.42	1.52 (0.91–2.53)
Number of medication-related problems				0.007		
0	179 (11.7)	144 (12.6)	35 (8.9)	0.01	−0.64	1.90 (1.16–3.10)
1	288 (18.8)	232 (20.3)	56 (14.3)	<0.001	−0.52	2.10 (1.38–3.18)
2	294 (19.2)	216 (18.9)	78 (19.9)	0.11	−0.49	1.37 (0.93–2.03)
3	220 (14.4)	161 (14.1)	59 (15.1)	0.47	−0.32	1.17 (0.77–1.77)
4	176 (11.5)	122 (10.7)	54 (13.8)	0.57	0.10	1.13 (0.74–1.75)
≥5 (reference)	376 (24.5)	266 (23.3)	110 (28.1)			
Number of prescribers				0.66		
1–5 (reference)	743 (48.5)	536 (47.0)	207 (52.8)			
6–10	635 (41.4)	479 (42.0)	156 (39.8)	0.55	0.09	1.089 (0.82–1.44)
11–15	131 (8.5)	106 (9.3)	25 (6.4)	0.29	0.29	1.329 (0.78–2.26)
≥16	24 (1.6)	20 (1.8)	4 (1.0)	0.44	0.47	1.592 (0.49–5.21)
Number of pharmacies				0.92		
1 (reference)	450 (29.4)	336 (29.4)	114 (29.1)			
2	449 (29.3)	336 (29.4)	113 (28.8)	0.95	−0.01	0.99 (0.71–1.37)
3	322 (21.0)	244 (21.4)	78 (19.9)	0.63	0.09	1.10 (0.76–1.58)
≥4	312 (20.4)	225 (19.7)	87 (22.2)	0.82	−0.04	0.96 (0.66–1.39)
Depression						
Yes	259 (16.9)	197 (17.3)	62 (15.8)	0.41	0.15	
No (reference)	1274 (83.1)	944 (82.7)	330 (84.2)			
Number of controlled medication classes				0.001		
1 (reference)	339 (22.1)	228 (20.0)	111 (28.3)			
2	598 (39.0)	380 (33.3)	218 (55.6)	0.15	−0.22	0.81 (0.60–1.09)
3	480 (31.3)	423 (37.1)	57 (14.5)	<0.001	1.06	2.89 (1.99–4.21)
≥4	116 (7.6)	110 (9.6)	6 (1.5)	<0.001	1.86	6.44 (2.68–15.46)
Number of oral corticosteroid fills				0.53		
0 (reference)	1040 (67.8)	767 (67.2)	273 (69.6)			
1	264 (17.2)	194 (17.0)	71 (18.1)	0.35	−0.16	0.85 (0.61–1.20)
2	98 (6.4)	72 (6.3)	26 (6.6)	0.35	−0.25	0.78 (0.46–1.31)
≥3	131 (8.5)	108 (9.5)	22 (5.6)	0.54	0.17	1.18 (0.69–2.03)
Number of albuterol inhalers						
≤2 (reference)	647 (42.2)	414 (36.3)	233 (59.4)			
≥3	886 (57.8)	727 (63.7)	159 (40.6)	<0.001	0.81	2.25 (1.74–2.91)
90-day supply of inhalers						
No (reference)	1088 (71.0)	759 (66.5)	329 (83.9)			
Yes	445 (29.0)	382 (33.5)	63 (16.1)	<0.001	0.95	2.57 (1.88–3.53)

Note. Bonferroni-adjusted *p* value = 0.0045.

**Table 2 jcm-11-06985-t002:** Results from logistic regression evaluating the association between patient characteristics and adherence to controller medications in 2020 (N = 1533).

Characteristic	Overall Cohort	With High Adherence	With Non-Adherence	*p* Value	B Value	Adjusted Odds Ratio
	N = 1533	N = 891 (58.1)	N = 642 (42.9)			(95% confidence interval)
	N (%)	N (%)	N (%)			
Age, year				0.88		
65–74 (reference)	644 (42.0)	371 (41.6)	273 (42.5)			
75–84	656 (42.8)	401 (45.0)	255 (39.7)	0.77	0.05	1.05 (0.76–1.45)
≥85	233 (15.2)	119 (13.4)	114 (17.8)	0.79	−0.1	0.94 (0.61–1.46)
Sex						
Male (reference)	629 (41.0)	367 (41.2)	262 (40.8)			
Female	904 (59.0)	524 (58.8)	380 (59.2)	0.25	0.18	1.19 (0.88–1.62)
Number of medications				0.47		
8–10 (reference)	456 (29.7)	250 (28.1)	206 (32.1)			
11–13	400 (26.1)	235 (26.4)	165 (25.7)	0.56	0.12	1.12 (0.76–1.66)
14–16	299 (19.5)	181 (20.3)	118 (18.4)	0.83	−0.05	0.95 (0.62–1.47)
17–19	200 (13.0)	113 (12.7)	87 (13.6)	0.20	−0.32	0.72 (0.44–1.18)
≥20	178 (11.6)	112 (12.6)	66 (10.3)	0.62	0.14	1.15 (0.66–2.01)
Number of medication-related problems				0.03		
0	179 (11.7)	117 (13.1)	62 (9.7)	0.50	−0.20	1.22 (0.68–2.21)
1	288 (18.8)	181 (20.3)	107 (16.7)	0.35	0.21	0.80 (0.50–1.28)
2	294 (19.2)	154 (17.3)	140 (21.8)	0.08	−0.52	0.66 (0.41–1.05)
3	220 (14.4)	123 (13.8)	97 (15.1)	0.19	−0.62	0.73 (0.45–1.18)
4	176 (11.5)	104 (11.7)	72 (11.2)	0.14	−0.42	1.51 (0.87–2.63)
≥5 (reference)	376 (24.5)	212 (23.8)	164 (25.5)			
Number of prescribers				0.13		
1–5 (reference)	874 (57.0)	475 (53.3)	399 (62.1)			
6–10	544 (35.5)	335 (37.6)	209 (32.6)	0.80	−0.04	0.96 (0.70–1.32)
11–15	98 (6.4)	72 (8.1)	26 (4.0)	0.61	0.17	1.18 (0.63–2.22)
≥16	17 (1.1)	9 (1.0)	8 (1.2)	0.02	−1.35	0.26 (0.08–0.83)
Number of pharmacies				0.08		
1 (reference)	450 (29.4)	254 (28.5)	196 (30.5)			
2	449 (29.3)	271 (30.4)	178 (27.7)	0.05	0.38	1.46 (1.00–2.14)
3	322 (21.0)	180 (20.2)	142 (22.1)	0.87	0.03	1.03 (0.69–1.55)
≥4	312 (20.4)	186 (20.9)	126 (19.6)	0.05	0.44	1.56 (1.01–2.41)
Depression						
Yes	259 (16.9)	152 (17.1)	107 (16.7)	0.38	0.19	1.21 (0.79–1.85)
No (reference)	1274 (83.1)	739 (82.9)	535 (83.3)			
Number of controlled medication classes				<0.001		
0–1 (reference)	595 (38.8)	175 (19.6)	420 (65.4)			
2	432 (28.2)	272 (30.5)	160 (24.9)	0.18	−0.25	0.78 (0.54 -1.12)
3	397 (25.9)	340 (38.2)	57 (8.9)	<0.001	0.82	2.28 (1.50–3.47)
≥4	109 (7.1)	104 (11.7)	5 (0.8)	<0.001	1.88	6.55 (2.47–17.35)
Number of oral corticosteroids fills				0.57		
0 (reference)	1139 (74.3)	641 (71.9)	498 (77.6)			
1	187 (12.2)	115 (12.9)	72 (11.2)	0.21	−0.29	0.75 (0.48–1.18)
2	85 (5.5)	55 (6.2)	30 (4.7)	0.60	0.20	1.22 (0.59–2.51)
≥3	122 (8.0)	80 (9.0)	42 (6.5)	0.91	−0.03	0.97 (0.56–1.68)
Number of albuterol inhalers						
<3 (reference)	729 (47.6)	324 (36.4)	405 (63.1)			
≥3	804 (52.4)	567 (63.6)	237 (36.9)	0.01	0.39	1.48 (1.09–1.99)
90-day supply of inhalers						
No (reference)	1025 (66.9)	509 (57.1)	516 (80.4)			
Yes	508 (33.1)	382 (42.9)	126 (19.6)	<0.001	0.89	2.44 (1.73–3.46)
High adherence to any inhalers in 2019						
No (reference)	392 (25.6)	155 (17.4)	237 (36.9)			
Yes	1141 (74.4)	736 (82.6)	405 (63.1)	<0.001	0.81	2.24 (1.63–3.08)

Note. Bonferroni-adjusted *p* value = 0.0042.

## Data Availability

Not applicable.

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
