# Peer review of "Medication Adherence in Medicare-Enrolled Older Adults with Chronic Obstructive Pulmonary Disease before and during the COVID-19 Pandemic"

_jcm, 2022, doi:10.3390/jcm11236985_

Round 1

Reviewer 1 Report

This is a straightforward analysis of adherence to inhaled medication before versus during the COVID pandemic.

A few items deserve some further discussion.

1. Yes or no adherent is set at more or less than 80%. This is an international way to express adherence but histograms would be more informative.

2. What is the usual variation in PDC in the years before COVID?

3. How was adherence specifically for inhaled corticosteroids?

4. Is it possible that patients were less active during the pandemic and therefore needed less reliever therapy?

Author Response

Point 1: Yes or no adherent is set at more or less than 80%. This is an international way to express adherence, but histograms would be more informative.

Response 1: Thank you for your suggestion. Histograms have been added to express the PDC, the proportion of patients with high adherence to their inhalers, and the proportion of patients with at least one combination inhaler in 2019 and 2020. 

Point 2: What is the usual variation in PDC in the years before COVID?

Response 2: We only measured the PDC during Jan 1-July 31, 2019, and Jan 1-July 31, 2020 periods. We used similar periods of time in case adherence changes based on the season. We did not have access to adherence data prior to 2019 and therefore cannot account for historical trends.  We added a statement reflecting this in the limitations. 

Point 3: How was adherence specifically for inhaled corticosteroids?

Response 3: Sorry if there was any confusion. This information was originally presented in Table 2. It can now be found in Figure 2 and the Result section of the manuscript in Lines 122-126.

Point 4: Is it possible that patients were less active during the pandemic and therefore needed less reliever therapy?

Response 4: Thank you so much for your insightful suggestions. We found fewer patients received more than three relievers during the pandemic in this study. It may have been due to patients being less active during the pandemic. We added this statement in the Discussion section of our manuscript.

Point 5: The abstract is good; however, the authors do not mention methods how the adherence to inhaled medications was measured to calculate Proportion of Days Covered e.g. self-reported, pharmacy record-based, etc. This is an important issue and potential limitation of the study."

Response 5: We are sorry about the confusion; this is noted in the Data Sources section in the Methods in Line 82. The PDC was calculated based on the pharmacy claim data.

Reviewer 2 Report

The manuscript describes the statistical analysis of the medication adherence of older patients with COPD before and during the COVID-19 pandemic using Medicare insurance records.

My main concern is the methodology of the logistic regression modeling.

Specifically, the logistic regression performance is missing. It is necessary to report the appropriate model performance estimates, e.g accuracy, F1 score, ROC AUC (plot and score) to analyze the importance/significance of the features. If the performance is low, other models can be explored (elastic net, random forest, etc)

I don’t see a training/validation set split into the methods: model performance should be reported for the validation set. Was the dataset imbalanced: what is the proportion of cases and controls in each model? Use the appropriate metrics in case the dataset is imbalanced.

It will help to report variables' coefficients along with the p-values, maybe plot the variables with the top coefficients

Author Response

Point 1: The manuscript describes the statistical analysis of the medication adherence of older patients with COPD before and during the COVID-19 pandemic using Medicare insurance records.

My main concern is the methodology of the logistic regression modeling.

Specifically, the logistic regression performance is missing. It is necessary to report the appropriate model performance estimates, e.g accuracy, F1 score, ROC AUC (plot and score) to analyze the importance/significance of the features. If the performance is low, other models can be explored (elastic net, random forest, etc)

I don’t see a training/validation set split into the methods: model performance should be reported for the validation set. Was the dataset imbalanced: what is the proportion of cases and controls in each model? Use the appropriate metrics in case the dataset is imbalanced.

It will help to report variables' coefficients along with the p-values, maybe plot the variables with the top coefficients.

Response 1: Authors appreciate the reviewer’s questions about model fit and ability to predict future high adherence. The Logistic regression we performed was an exploratory analysis to determine if there were any variables associated with having high adherence while maintaining other variables as constants. Authors did not intend to create a predictive model (which could be assessed for model fit and predict high adherence). Our objective was to identify the most important variables associated with high adherence in patients with COPD, as we have reported in the manuscript. Authors believe a proper predictive model would require access to time series data reflecting healthcare utilization and medication costs data we unfortunately do not have access to.
